# Public Health Safety in Community Living Circles Based on a Behavioral Motivation Perspective: Theoretical Framework and Evaluation System

**DOI:** 10.3390/bs13010026

**Published:** 2022-12-27

**Authors:** Qikang Zhong, Bo Li, Yue Chen, Jiawei Zhu

**Affiliations:** 1School of Architecture and Art, Central South University, Changsha 410083, China; 2School of Geosciences and Info-Physics, Central South University, Changsha 410083, China

**Keywords:** community living circle, public health safety, daily life activities, behavioral motivation, healthy communities

## Abstract

**Highlights:**

A theoretical framework for the research of public health safety in community living circles is based on a behavioral motivation perspective.

**What are the main findings?**
Building a “geographical environment–life activity–public health” model for evaluating health behavior.Building a public health safety evaluation system for community living circles.

**What is the implication of the main finding?**
A new approach to the theory of “healthy communities” based on direct impact factors.The findings can be used in community living circles for “health checks”, and the planning of healthy and safe living circles.

**Abstract:**

Public health problems, such as the spread of COVID-19 and chronic diseases, are mainly caused by the daily life activities of community residents. Therefore, there is a need to build a healthy and safe community living circle through the evaluation of health behaviors in daily life. This paper proposes a theoretical framework and evaluation system for public health safety in community living circles, from a behavioral motivation perspective. Firstly, based on the behavioral motivation theory, a theoretical framework for the study of public health safety in community living circles is constructed from the perspective of the “project–activity–health” coupling relationship network, regarding community residents’ daily life activities. Then, a public health safety evaluation system for community living circles is proposed based on this framework, which includes the following: (1) identifying the scope of community living circles based on Spatio-temporal Activities Analysis; (2) Based on the theory of protection motivation, a health behavior evaluation model based on the three elements of “spatial and temporal geographical environment–daily life activities–public health safety” is established; (3) Based on the hierarchy of public health problems, a public health safety evaluation model of the community living circle is established. The behavioral motivation-based evaluation system explores a new approach and research paradigm for community-scale public health safety theory; this will help to achieve the goal of “healthy communities” when further empirical evidence is available.

## 1. Introduction

The daily life activity of community residents is a direct factor that causes public health problems, such as infectious diseases and chronic diseases [1,2,3]. Statistics from the World Health Organization (2018) show that non-communicable diseases, caused by inappropriate daily life activities, cause more than 41 million deaths each year, accounting for 71% of global deaths. The Chinese academicians, Longde Wang (2008) and Nanshan Zhong (2015), pointed out that most of the diseases caused by living habits and behaviors can be prevented by adjusting lifestyles [4,5]. At present, daily life activities not only affect the prevention and control of the COVID-19 epidemic and public health, but are also studied by spatiotemporal geographers and urban planners in the spatial planning of residential areas [6,7]. The “community life circle (CLC)”, which is defined by the temporal and spatial scope of the daily activities of community residents, has been listed as the basic planning unit of the “Planning and Design Standards for Urban Residential Areas” (2018). How to integrate health into CLC [8], and construct healthy or safe CLCs [9,10], has become an issue of high concern for the relevant disciplines, and needs to be solved urgently.

Daily life activities in the community directly affect public health safety, and the spatiotemporal and geographical elements, such as the built environment, indirectly affect public health by affecting life behaviors [11]. However, most of the existing studies analyze public health problems from the perspective of indirect influencing factors, such as community resources and the environment [12,13,14,15,16]. From the perspectives of epidemiology, pathology, and psychology, scholars in the fields of health geography and medical geography have studied the impact of the spatial construction of community health care resources and the behavior of community residents on health [17,18,19]. Scholars in the field of urban geography and urban-rural planning mainly focus on the relationship between the built environment of the community and the health of the residents. The research content mainly involves a balanced diet, physical activity, timely medical treatment, and the completeness and accessibility of related facilities [20,21,22,23]. In addition, some scholars have studied the influencing factors of community health behaviors [24,25,26], as well as the influence of certain life behaviors on certain diseases [27,28,29]. However, the systematic theoretical framework of daily life activity in the community and public health issues still needs to be further explored.

The evaluation system of community public health safety also needs to be improved. Existing scholarly research has proposed a measurement index system for healthy communities [30]. In 2020, the China Urban Science Research Association also released the “Healthy Community Evaluation Criteria”, which proposed the seven indicators of air, water, comfort, fitness, humanities, service, and innovation [31]. These index systems are not based on the daily life activities of residents, but are established directly from the social environment and physical environment. Therefore, the evaluation objects are still the resources and environmental factors that indirectly affect health. On the other hand, in the evaluation research of community health behavior, the important influence of CLC environment on health behavior is often ignored, and the evaluation model is established; here, individual psychological cognitive factors are the main variables, such as behavioral beliefs [32], behavioral attitudes [33], personal values [34], risk perception [35], interpersonal relationships [36], etc.

Therefore, from the perspective of behavioral motivation, this paper proposes a theoretical framework for the study of daily life activity in the community and public health safety, and clarifies the “planning-activity-health” coupling network of the CLC. In addition, we have established a health behavior evaluation model for the three elements of “space-time geographic environment–daily life activities–public health safety”, and constructed a public health safety evaluation system for the community life circle. It aims to provide a new way of thinking and a new research paradigm for the construction of healthy communities.

## 2. Research Theory of Community Life Behavior Motivation and Public Health Safety

### 2.1. Behavioral Motivation Theory and CLC Public Health Safety

The behavioral motivation theory emphasizes the study of behavior as a function of the total physical and social situation. Behavioral motivation refers to the subjective desire and intention of the behavior subject to achieve a certain goal [37]. Spatiotemporal geography and health psychology have introduced the concept of “behavioral motivation” into the research related to daily life activities; this is in order to solve the relationship between the subjective behavior intention and the behavioral consequences [38,39,40,41]. Scholars of spatiotemporal geography propounded the “project–activity” system theory, in order to analyze the behavioral mechanism of the subjective and objective interaction process. They believe that a project is a series of necessary simple or complex tasks to accomplish any wishes and goals, forming a social ecological closed-loop decision-making process in the form of “project–activity–project” [41,42,43,44]. The daily activities and project arrangements of individuals, families, and community organizations, as well as the local order space formed by the interaction of different subjects, are important components of the CLC [45,46,47,48]. Health psychologists have propounded the Protection Motivation Theory (PMT). Firstly, this theory is used to study whether the protection motivations formed by people’s fear appraisal regarding risk information will affect their self-protection behaviors, including the constructs of severity, vulnerability, and response-efficacy; later added is the self-efficacy to shape the current Protection Motivation Theory. Therefore, the PMT is used to illustrate how people seek self-protection (intention) from a harmful or stressful life [49,50]. Through a large number of applied studies in the fields of health promotion and life behavior changes, the theory has become a health promotion method and has solved many public health problems [51,52,53,54]. Based on the above, in this research, we choose PMT as the theoretical basis to explain the cognitive process and interaction mechanism of the residents’ healthy behaviors in the CLC. This research is to predict the behavior change process of individual residents from the perspective of motivational factors, and to test how individuals will respond in a favorable or unfavorable way when faced with certain risk factors in the internal and external environment [55,56,57,58].

Combining the above behavioral motivation theories, we can better understand the complexity of healthy behaviors in daily life, and more scientifically study the relationship between activity behaviors planned in daily life and public health problems. This protective motivation to establish healthy behaviors, in turn, affects the adjustment and decision-making process of project behaviors, and finally forms a closed-loop community health behavior evaluation system in daily life. Therefore, from the perspective of behavioral motivation, the public health safety of the CLC can essentially be regarded as the coverage of the community’s daily life health behaviors in time and space. Its boundaries, forms, internal components, and service facilities all maintain and support healthy behaviors, thereby ensuring the public health safety of the CLC.

### 2.2. Types and Relationships between Community Daily Life Activities and Public Health Problems

Daily life activities refer to the activities that residents perform in certain ways to achieve their desired goals, in order to meet the needs of daily life [59,60]. According to different types of daily life projects, Gehl divides outdoor space activities into necessary activities, spontaneous activities, and social activities (see Table 1) [61]. Among them, necessary activities refer to activities that residents must do for their own survival, including eating, sleeping, going to school, shopping, etc. Optimal activities refer to the willingness of residents to participate in activities, and the external environment has a greater impact on such activities. In general, residents carry out spontaneous activities under suitable temperature and good space conditions, including exercise, walking, entertainment, recreation, and sun exposure. Compared with spontaneous activities, social activities take place in public spaces, rely on the participation of others, and cannot be completed independently, including chatting, greeting, and various public communication activities [62].

With reference to the World Health Organization classification basis, the International Classification of Diseases, and the professional assessment classification guidelines [63,64,65], the public health problems in the CLC can be divided into five categories. The first category is infectious diseases, including respiratory, body surface and digestive system infectious diseases; the second category is chronic non-communicable diseases, including cardiovascular diseases, diabetes, and chronic systemic respiratory diseases; the third category is accidental injuries, including traffic accident injury, fall collision injury and drowning; the fourth category is bad health behavior, including irregular diet and personal addiction; the fifth category is mental and mental health diseases, including depression and Alzheimer’s disease.

The “project–activity” system in the daily life of the community has a direct impact on the public health problems of the CLC [66]. Different residents’ daily activities lead to different public health problems, such as daily food and housing activities that can lead to obesity, diabetes, and further chronic diseases [67,68,69]. Daily wellness activities can effectively prevent cancer, cardiovascular disease, and diabetes [28,70,71,72,73]. Appropriate walking physical activity also reduces the incidence of accidental falls in elderly patients [74] and promotes changes in adverse behaviors [75,76]. Therefore, the coupling relationship between the daily life activity of community residents and public health issues is the theoretical basis for carrying out related research.

### 2.3. Theoretical Framework of CLC Public Health Safety Research

Based on the impact mechanism of daily life activities on health and safety, clarifying the “project–activity–health” relationship network is the basic premise and necessary way to realize the CLC public health safety research (Figure 1). The behavioral project plays a role in promoting residents’ daily activities or regulating the frequency reduction through subjective motivations, such as activity goals and meanings. Based on the interaction between daily life activities and public health problems, we can find the factors of residents’ life behavior that increase the risk of certain diseases, analyze the ability to change activity behavior in response to the health risk, and then explain, predict, and promote the establishment of health behavior. This protective motivation to establish healthy behaviors generates a behavioral project awareness of positive implementation and execution, or negative adjustment and improvement, which in turn affects the adjustment and decision-making process of project behaviors. Therefore, only by studying the ‘project–activity–health’ system as a whole can we more accurately assess the public health impact of community residents’ daily life activities and carry out the public health safety research of the CLC.

In summary, this research constructs a theoretical framework for the CLC public health safety research based on the “project–activity–health” system (Figure 2) and the Influence Structure of Daily Life Activities and Public Health Problems Based on Protection Motivation Theory (Figure 3). The research object is composed of daily life activities in the community, public health safety, and the environment of the community life circle. Among them, lines with the same color represent the significant effect of the same activity on public health problems, and the dotted or solid state of lines denotes the strength of the significant effect. The environment of the community life circle mainly consists of the built environment and the social environment. The construction of this built environment system is based on the traditional 5D built environment framework [21,77]. The social environment is mainly composed of economic, demographic, culture, policy, and other elements [78,79,80,81]. The Protection Motivation Theory in Figure 3 is mainly divided into threat assessment and response assessment. Threat assessment consists of four elements: vulnerability, severity, internal reward, and external reward. Response assessment consists of three elements: response efficiency, self-efficacy, and cost of response. Severity refers to the threat that the risk factors may cause to the individual’s own interests; Vulnerability refers to an individual’s perceived likelihood of being threatened by such a risk factor or suffering from such a disease; Internal reward refers to experiencing the self-satisfaction or internal positive feeling when an individual is immersed in a specific activity or behavior; External reward refers to the factors from peers, families or other sociogroups that can strengthen individual behaviors; Response efficiency refers to the individual’s cognition of whether a certain protective behavior is effective; Self-efficacy refers to an individual’s cognition of his/her ability to take certain protective behaviors.

## 3. CLC Health Behavior Evaluation Method

### 3.1. Evaluation Theory and Precondition

On the basis of the research theoretical framework, a protection motivation model can be established to evaluate the residents’ health behavior. Taking the health protection motivation of behavior as the starting point, its logical relationship with spatio-temporal geographical environment elements and daily life behavior elements is analyzed. The model framework of the Protection Motivation Theory is divided into information source, the cognitive mediation process, and the response mode. Firstly, information sources include personal internal factors and the geographical environment of the community life circle, both of which have an impact on individual cognitive processes. Secondly, the cognitive mediation process is the core part of the theory, including threat assessment and response assessment. Threat assessment includes severity, vulnerability, internal reward, and external reward, which is the cognition of risk factors. Response assessment includes response efficacy, self-efficacy, and response cost, which is the cognition of the health threat processing ability. Lastly, the response mode refers to the behavior that individuals ultimately adopt after evaluation, including negative health risk behaviors (such as smoking, drinking, poor diet, etc.) and positive health behaviors (such as regular exercise, regular physical examination, etc.), and compliance behaviors. The Protection Motivation Theory can scientifically and reasonably explain the mechanism of project change [82,83,84,85,86,87], so that researchers can more clearly understand the cognitive process and interaction mechanism of the health behavior, thus laying a solid foundation for further public health safety interventions.

Both the <Technical Guidelines for Community Life Circle Planning>, issued by the Ministry of Natural Resources of China, and the <Urban Master Planning of Shanghai (2017–2035)>, issued by the Shanghai municipal government, propose that the 15-min walkable range should be taken as the spatial scale of the CLC to allocate various functions and facilities necessary for residents’ basic living. For this, the division of the CLC’s space is the precondition of the health evaluation of daily life activities in the community. Most of the existing division methods define the CLC’s spatial range as the accessible range for residents walking for 15 min. However, the coverage is only the expression of geometric figures, which does not reflect the real-life trajectory of residents and the differentiated resident activities in different CLCs. In order to reflect the real scope of residents’ daily lives, it is necessary to first use the 15-min walkable range as the initial range, and then use wearable device data to identify residents’ travel stops and count the frequency of visits [88,89,90,91,92]. The intersection of the minimum convex hull of the residents’ travel stop points and the walking circle is the daily activity range circle (Figure 4).

### 3.2. Health Behavior Evaluation Model of CLC

Within the scope of the CLC, based on the Protection Motivation Theory, the model is established separately for each type of daily life activity; the health problems related to this activity type are extracted from the ‘project–activity–health’ relationship network and are integrated into the corresponding evaluation model, which is divided into three steps to construct health behavior (Figure 5).

The first step is to conduct a questionnaire survey of residents in the study area. The questionnaire includes three parts: general situation, implementation of the health behavior, and protection motivation factors related to each type of health problem. Among them, the intention to perform health behaviors and the measurement items of protection motivation factors adopt the five-point Likert scale. Corresponding measurement items were designed for each variable (severity, vulnerability, self-efficacy, response efficacy, response costs, healthy behavioral intention, etc.). The Likert scale gives a score for each answer, with 1 to 5 points from ‘very agree’ to ‘very disagree’. The second step is to set the variable system of the evaluation model (Table 2). Population control variables, protection motivation variables, and health behavior dependent variables were derived from the questionnaire. At the same time, the corresponding environment variables are obtained in the identified CLC range, and the influence of spatio-temporal geographical environment elements and social environment elements are added to the model. In order to ensure the reliability and robustness of the model, it is necessary to verify the reliability and validity of the questionnaire according to the activity category. According to the clonal Bach coefficient (>0.6), KMO test (>0.6), and Bartlett spherical test (<0.05), the variables used for modeling are selected. Meanwhile, the population factors, environmental factors, and intentions to daily life activity were analyzed by single factor analysis, leaving statistically significant variables (*p* < 0.05).

The third step is to establish a health behavior evaluation model for each type of project activity. Considering that the dependent variables are ordinal categorical variables, the basic model prefers ordinal logistic regression, but if the parallelism hypothesis is violated, the generalized logistic regression is used to iterate the final model according to the goodness of fit test of the model. On the basis of this model, the marginal effect research can be carried out, that is, when other variables remain unchanged, the influence of the change of an independent variable on healthy behavioral intention can be studied. The model can determine variables that have a significant impact on the daily life behavioral intention, based on health and safety considerations (such as geographical environment elements and social environment elements, internal rewards, or self-efficacy). If the final response assessment is health risk behavior, health education and intervention strategies are provided to the target population, so that their behavioral project can be adjusted and improved until they achieve positive health behavior (Figure 6). Finally, the CLC health behavior evaluation model and response mechanism, based on the three elements of “space-time geographical environment-daily life activities-public health safety”, can be established.

## 4. Construction of CLC Public Health Safety Evaluation System

### 4.1. Public Health Safety Evaluation Steps of CLC

The indicators in the CLC public health safety evaluation system need to be able to reflect the characteristics and contents of the health community evaluation index system scientifically and reasonably. At the same time, each evaluation index should be operable, comparable, and can be investigated through certain methods and approaches. Based on the theoretical framework of community life behavior motivation and public health safety research, a hierarchical structure of public health problems can be established to establish the relationship among health problems, daily behavior, and the community life circle environment. First, set up sub-questions regarding the CLC public health safety, according to the eight types of daily life activities in the ‘plan–activity’ system. Then, according to the five steps, the following process is observed (Figure 7): (1) The variable coefficients that significantly affect the healthy behavioral intention of sub-questions are extracted. (2) Subjective and objective weight assignments are carried out on each variable of sub-questions, through the Analytic Hierarchy Process and the Entropy Weight Method. (3) An evaluation model of healthy behaviors is constructed and the health scores of the CLC sub-questions are obtained using the Synthetical Index Method. (4) The public health safety evaluation system model of the CLC is constructed; additionally, the Analytic Hierarchy Process (scored by experts) is used to weight and synthesize the health scores of various sub-questions, and the synthetical index of the community health life circle will be finally obtained. (5) Division of evaluation criteria: based on the requirements of practicality and advancement, with references to relevant studies, the synthetical index of the community health life circle is divided into grades using the Equal Interval Method or Natural Discontinuity Method. The larger the synthetical index is, the better the health development of the CLC will be. The model formula is:(1)$$h=\sum_{i=1}^{n}y_{ij}\times w_{i}\times z_{i}$$
(2)$$H=\sum^{l=1}h\times d_{l}$$

Note: H is the synthetical index of community health life circle; h is the health score of sub-questions; $y_{ij}$ is the standardized values of variables data; $w_{i}$ is the weight of each variable; $z_{i}$ is the variable coefficient that significantly affects the healthy behavioral intentions of the sub-questions; n is index for evaluation; l is the sample for daily life activities evaluation; i is the evaluation sample; *j* is the index value.

### 4.2. Public Health Safety Evaluation System of CLC

After completing the sub-questions of the health scoring model, it is necessary to quantify the importance of sub-questions in health and safety issues. Using the analytic hierarchy process, which is a combination of definiteness and quantitative, and the systematic analysis method, the weight from the sub-questions to the core problem is obtained. In view of the fact that there are more qualitative judgments than quantitative analyses in this process, the credibility of the weight of each sub-question can be increased through expert evaluation in combination with the Delphi method. Then, the pairwise comparison matrix calculates the maximum eigenvalue and its corresponding eigenvector. Then, the maximum eigenvalue and its corresponding eigenvector are computed via a pairwise comparison matrix. If the consistency test does not pass, the pairwise comparison matrix needs to be reconstructed. Otherwise, the feature vector is normalized as the weight vector, and the public health safety evaluation system model of the CLC is obtained. Using the evaluation system model to carry out empirical research, the comprehensive and systematic ‘health examination’ of the CLC in the study area can be carried out to verify the scientificity and feasibility of the evaluation system (Figure 8).

## 5. Conclusions

Different from the existing research ideas and methods, this paper takes the daily life activities of community residents as the evaluation object, from the perspective of behavioral motivation, based on the Protection Motivation Theory of health psychology, establishes a health behavior evaluation model based on the logical chain of “space-time geographical environment–daily life activities–public health safety”, and constructs a public health safety evaluation system for the CLC. The system can not only assess the public health risks faced by the CLC, but also assess its ability to cope with risks; that is, the CLC has a controllable degree of resilience in the face of current risks. In the evaluation process, we must first explore the elements of residents’ life behavior that can increase the risk of health and safety problems, and then analyze the ability to change behavior in response to this health risk. Then, we must explain the establishment mechanism of the health behavior, and intervene in the health risk behavior in a targeted manner, so as to adjust or make decisions. Finally, a closed-loop reaction mechanism of “project–behavior–motivation evaluation feedback–project adjustment” is formed.

This research closely follows the development goal of the “Healthy Community Life Circle”, and establishes a public health safety evaluation system for the CLC, which can provide a scientific evaluation pattern for the urban communities of China to accomplish an optimum transformation. However, it has the following shortcomings: although this research is based on Gehl’s classification of outdoor activities, the detailed classification of daily life activities and the major activity motivations and objectives of it are based on the summary of the existing literature and the research needs, which still leaves something to be desired; limited by the temporal and spatial resolution, and the availability of data, this research only preliminarily determines the variables of the evaluation model, and the selection and definition of the variables still need to be improved. In addition, the theoretical framework and evaluation system constructed in this research remains to be further empirically investigated. Still, based on <“Healthy China 2030” Plan Outline>, <Guidelines on Comprehensively promoting the renovation of old housing estates>, and other relevant policies and regulations issued by the official government of China, it is feasible to apply our solution to the planning and design practice of the community health and safety life circle. It is expected that this research will help the implementation of the major decision-making deployment of the “Healthy China” strategy, and embed public health factors into urban planning and community governance. Ultimately, the goal of building a “healthy community” can be achieved.

## Figures and Tables

**Figure 1 behavsci-13-00026-f001:**
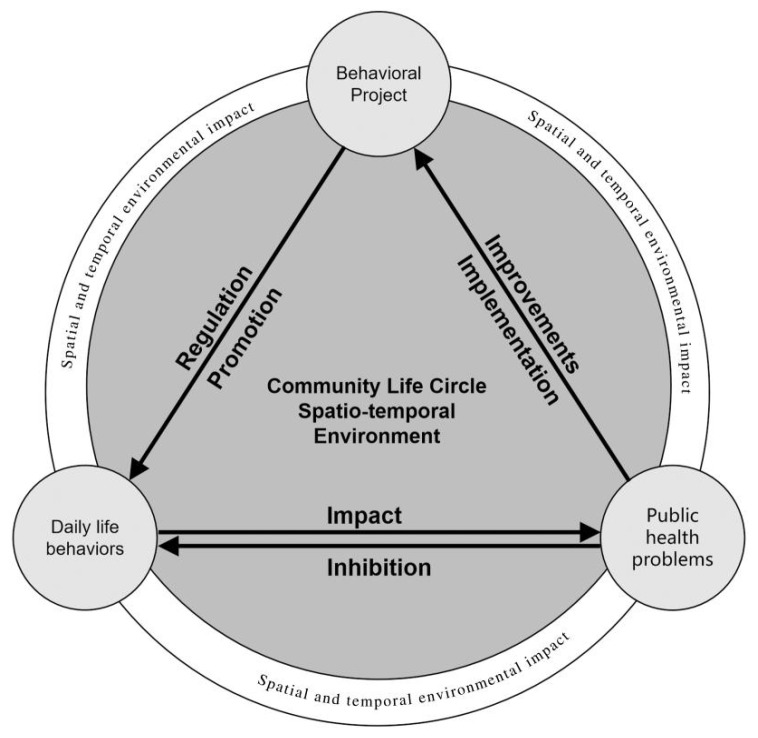
The “Project–Activities–Health” system of the CLC. Note: “public health problems” includes the illness that is being referred to in the text. “Behavioral Project” means the project which plays a role in promoting residents’ daily activities or regulating frequency reduction through subjective motivations, such as activity goals and meanings [41,42,43,44]. It is an intervention on one’s daily life activities. The community life circle environment promotes or inhibits residents’ daily life activities by influencing the residents’ daily life activities project, and then improves the health level through residents’ daily life activities or reversely adjusts residents’ behavioral projects through residents’ unhealthy behaviors.

**Figure 2 behavsci-13-00026-f002:**
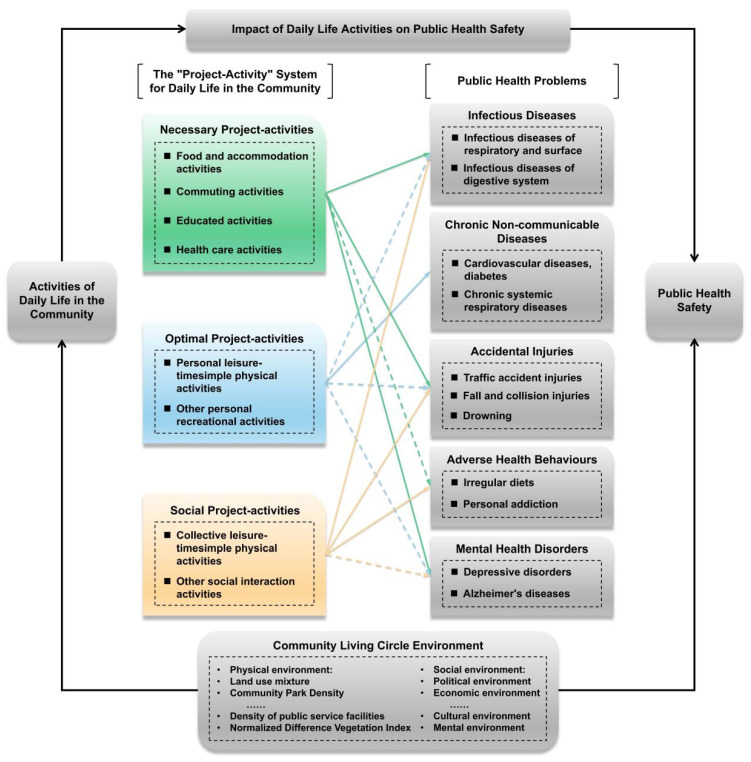
Theoretical framework for public health safety research in community life circles. Note: the diseases in color boxes under the “Public Health Problems categories” are examples, because we cannot list all the names of diseases.

**Figure 3 behavsci-13-00026-f003:**
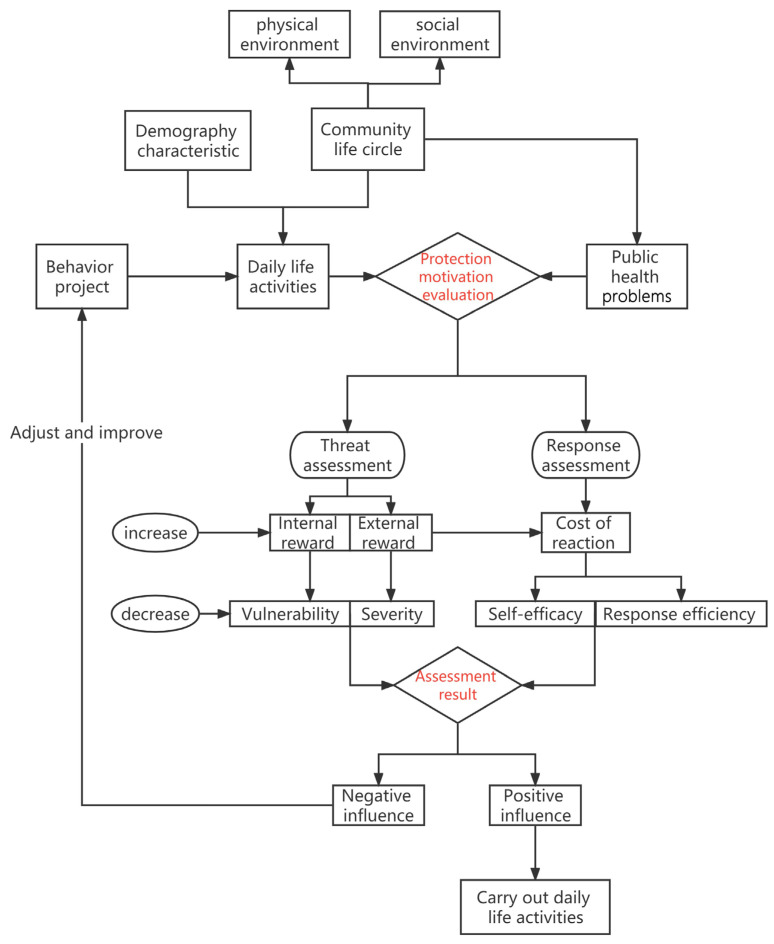
The Influence Structure of Daily Life Activities and Public Health Problems Based on Protection Motivation Theory. Note: "Protection motivation evaluation" refers to the health assessment of daily life activities and public health problems; “Assessment result” refers to the judgment of positive or negative influence on the health assessment of daily life activities and public health problems.

**Figure 4 behavsci-13-00026-f004:**
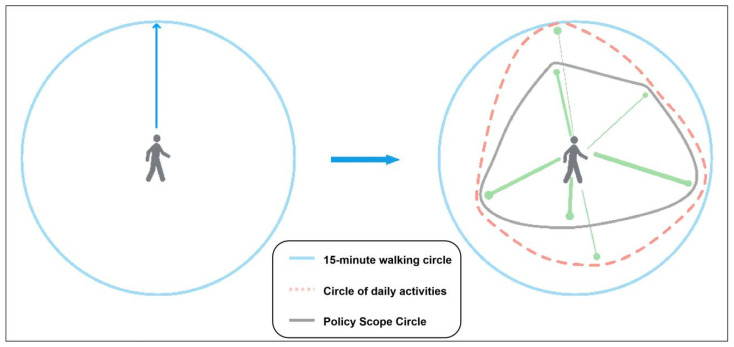
Schematic diagram of the community life circle scope identification.

**Figure 5 behavsci-13-00026-f005:**
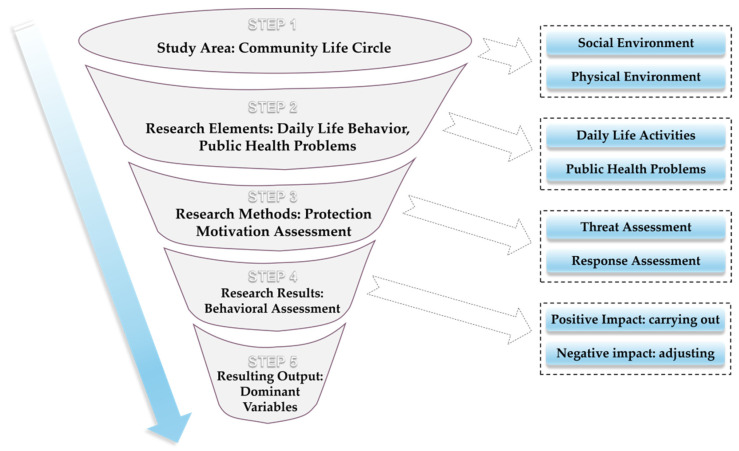
The overall framework of the health behavior evaluation model for community life circles. Note: “Dominant Variables” means the Independent variables that have significant effects on healthy behavioral intentions.

**Figure 6 behavsci-13-00026-f006:**
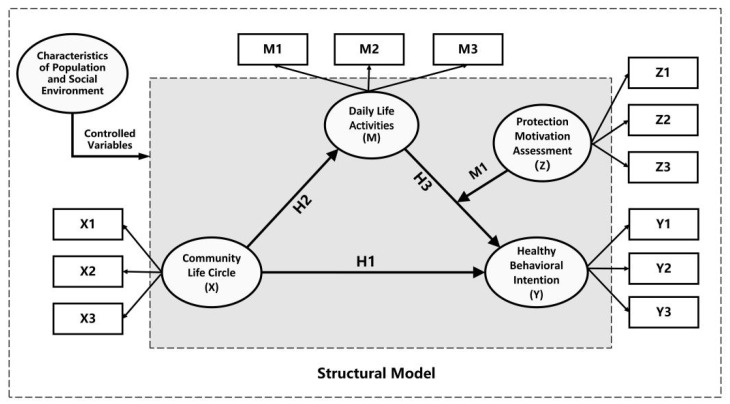
Health behavior assessment modeling structure for community life circles.

**Figure 7 behavsci-13-00026-f007:**
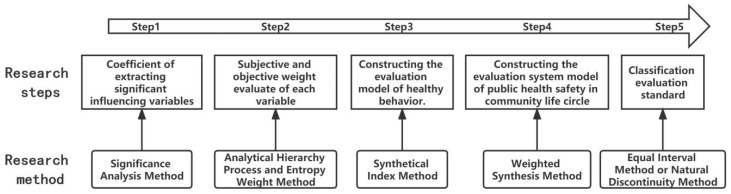
Steps in evaluating public health safety in community life circles.

**Figure 8 behavsci-13-00026-f008:**
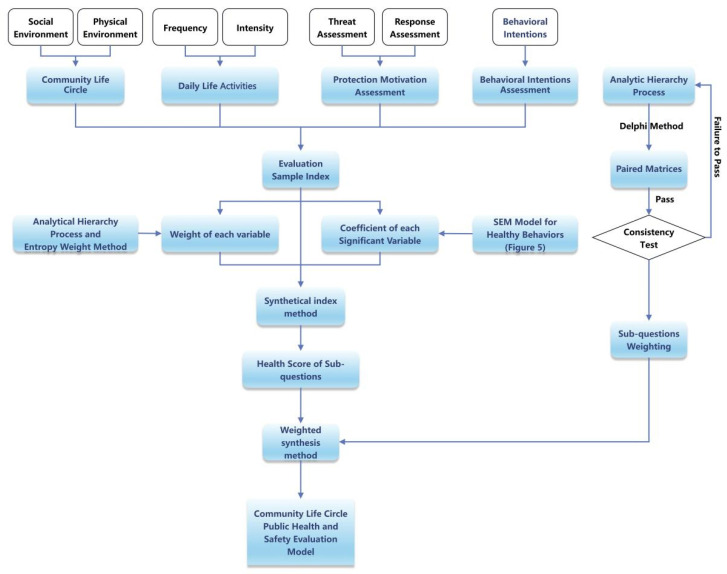
Public health safety evaluation system model for community life circles.

**Table 1 behavsci-13-00026-t001:** Classification system for daily life activities in the community based on the “Project–Activity” system.

Project Activities Categories	Categories of Activity Motivations and Objectives	Major Life Activities	Major Activity Motivations and Objectives
necessary activities	Activities that are less influenced by the external environment and are necessary to maintain one’s own survival or are required to be completed by the regulations.	Food and accommodation activities	Based on basic survival activities of feeding and housework
		Commuting activities	Daily commuting activities to and from work, or to and from the community
		Educated activities	Activities for groups of students going to and from school and receiving education
		Health care activities	Health and medical activities for middle-aged and elderly groups and the sick and infirm
spontaneous activities	Activities that are greatly influenced by the external environment and are undertaken spontaneously depending on one’s own will for the purpose of fitness, leisure and entertainment.	Personal leisure physical activity	Personal physical activity for the purpose of recreation and fitness
		Other personal recreational activities	The activities in which individuals pay more attention to the needs of their own spirit or enjoyment after meeting the needs of their basic physiological and security.
social activities	Social activities that are influenced by the external environment, for the purpose of fitness and recreation, and which are dependent on the participation of others and cannot be performed independently.	Group leisure and physical activity	Group physical activity for the purpose of recreation and fitness
		Other social interaction activities	Through various social activities, people can meet more different people to relieve the monotony and tedium of their life, exercise for their bodies and brains better, maintain vitality, delay senescence, and reduce the infestation of psychological diseases and senile diseases.

**Table 2 behavsci-13-00026-t002:** Evaluation model variable settings.

	Variables		Definition	Assignment
Dependent Variables	Behavioral Intention		The extent to which daily life activities affect their associated public health issues.	No impact = 1; Less impact = 2; General = 3; More impact = 4; Extraordinary impact = 5
Independent Variables	Density		Refers to the number of residents, housing units or employment in the unit area.	
	Diversity		Refers to the degree of land mix, i.e., different land use types and degree of complementarity in some area.	
	Destination Accessibility		Refers to the degree of proximity of non-residential land use (e.g., shops, parks, bus stops, etc.), mainly measuring the accessibility from one destination to another.	
	Design		Refers to the number of choices of directions and routes to destinations, which is often used to measure the difficulty of traveling between one place to another, and is usually shown by the street network pattern; The main measurements include block size, length, intersection density, complex network variables, etc.	
	Bus Stop Distance		Refers to the average value of the shortest route from the residence or workplace to the nearest bus stop, or uses bus route density, distance between bus stops, and the number of stops per unit area to quantify this indicator.	
Mediator Variables	Daily Life Activity		Frequency of daily life activity	
			Intensity of daily life activity.	
Moderator Variables	Threat assessment	Severity (Ser)	daily life activities may lead to public health diseases (Sev1).	Strongly disagree = 1; Disagree = 2; General = 3; Agree = 4; Strongly agree = 5
			Public health diseases increase the burden on families (Sev2).	
		Vulnerability (Vul)	daily life activities may cause diseases to spread to others (Vul1).	
			The possibility that daily life activities will cause public health diseases to disable oneself (Vul2).	
		Internal reward (Int)	Performing daily life activities makes residents feel very happy (Int1).	
			Performing daily life activities makes residents feel satisfied (Int2).	
		External reward (Ext)	daily life activities can attract the attention of the opposite sex (Ext1).	
			Performing daily life activities can be identified by others (Ext2).	
	Response assessment	Self-efficacy (Sel)	Residents are full of confidence that performing daily life activities can enhance their health (Sel1).	
			Residents are full of confidence in their self-restraint ability to inhint their daily life activities (Sel2).	
		Response efficiency (Res)	daily life activities can improve residents’ awareness of the benefits of disease prevention (Res1).	
			Controlling health risk behaviors can lead a comfortable life. Keeping healthy behaviors can improve the healthy level (Res2).	
		Cost of response (Cos)	Restraining daily life activities needs to bear additional burdens (Cos1).	
			Other obstacles or inconveniences may be encountered in the process of inhibiting daily life activities (Cos2).	
Controlled Variables for Demographic Characteristics	Gender (X1)			Male = 1; Female = 0
	Age (X2)		Actual Age	
	Educational Level (X3)			Never been to school = 1; Primary school = 2; Junior high school = 3; High school = 4; College degree or above = 5
	Marital Status (X4)			Unmarried = 1; Married = 0
	Career Information (X5)			
	Income (X6)			
Controlled Variables for Social Environment	Political Environment		Liberty of Speech (%)	No liberty = 1; Less liberty = 2; General = 3; More liberty = 4; Quite a lot of liberty = 5
			Citizens’ participation in politics (%)	Non-participation in politics = 1; Less political participation = 2; General = 3; More political participation = 4; Participation in politics very often = 5
			Policy Transparency (%)	Very opaque = 1; Opacity = 2; General = 3; Transparency = 4; Very transparent = 5
	Economic Circumstances		Net Income Per Capita	
			R&D Internal Appropriation Expenditure	
			Proportion of Secondary and Tertiary Industries in GDP	
			Regional GDP	
	Cultural and Educational Environment		Percentage of Cultural and Educational Activities of Community (%)	
			Average Annual Education and Publicity Expenses of Community	
			Per Capita Expenditure on Education	
	Psychological Environment		Community Identification	
			Community Satisfaction	
			Community Security	
			Community Participation	

Note: For example, the content of definition can be adjusted according to the specific sub-question.

## Data Availability

Not applicable.

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
