# Peer review of "Public Health Safety in Community Living Circles Based on a Behavioral Motivation Perspective: Theoretical Framework and Evaluation System"

_behavsci, 2022, doi:10.3390/bs13010026_

Round 1

Reviewer 1 Report (Previous Reviewer 1)

It is debatable whether achiving the goal  of building a ' healthy community ' is possible. Nevertheless, this is the conclusion ofthe authors based on the presented data.

Certainly the clarity of the manuscript is influenced by the graphs and figures.

In the conclusions, it is worth trying to determine the practicality of the solution, as well as the possibility of using it by other communities

Author Response

Dear reviewer,

Thank you very much for taking the time to review this manuscript. We appreciate your generous comments and suggestions! Please find my detailed response below and find my changes in the resubmission.

Yours sincerely,

Qikang Zhong, Bo Li, Yue Chen and Jiawei Zhu.

Reviewer 2 Report (New Reviewer)

Dear Authors and Editors,

This manuscript is scientifically sound, it presents very interesting approach, and theoretical framework for the research of public health safety in community living circles based on a behavioral motivation perspective. This topic will be of interest to the wide professional audience, from public health field of research and public health policy as well. Also, to other professionals engaged in community health research and programs, to health psychologists as well.

Here are some suggestions for improvements of the text:

At paragraph 2. / 2.1. sentence: "Health psychologists have put forward the theory of protective motivation…"  suggestion to use the same naming of the theory through the text, so it will be better to say: Health psychologists have put forward the behavioral motivation theory…

For the overall understanding and completeness of the article content, it will be beneficial to mention that (Lewin's)  behavioral motivation theory emphasised the study of behaviour as a function of the total physical and social situation.

The figures, images, and schemes are appropriate and clearly understand.

Table 1:  suggestion to divide sections by horizontal line in order to clearly see which statement in the last two columns belongs to which Category, what is now a bit unclear due to the amount of text and line spacing. (line dividing different Categories of Activity Motivation)

Author Response

Dear reviewer,

Thank you very much for taking the time to review this manuscript. We appreciate your generous comments and suggestions! Please find my detailed response below and find my changes in the resubmission.

Yours sincrely,

Qikang Zhong, Bo Li, Yue Chen and Jiawei Zhu.

Reviewer 3 Report (New Reviewer)

An interesting theoretical approach to the problem and a clearly outlined research project. A minor note: on page 10, some words are written in smaller font.

Author Response

Dear reviewer,

Thank you very much for taking the time to review this manuscript. We appreciate your generous comments and suggestions! Please find my detailed response below and find my changes in the resubmission.

Yours sincerely,

Qikang Zhong, Bo Li, Yue Chen and Jiawei Zhu.

This manuscript is a resubmission of an earlier submission. The following is a list of the peer review reports and author responses from that submission.

Round 1

Reviewer 1 Report

The article is interesting and innovative. I believe that the conclusions should be more specific.

For each article, the conclusions and their practical implementation are the most important

On the basis of such interesting material, I expect more precise observations and recommendations.

Whether it is possible to use of the results in other countries? Is an adaptation possible?

After taking into account the above comments, the work may be published

Author Response

Dear reviewer,

Thank you very much for taking the time to review this manuscript. We appreciate your generous comments and suggestions! Please find my detailed response below and find my changes in the resubmission.

Yours sincerely,

Qikang Zhong, Bo Li, Yue Chen and Jiawei Zhu.

Reviewer 2 Report

The authors have attempted to bring together elements of behavioural motivation, spatial and temporal geography and public health to propose a framework and evaluation system. While the ambition is to be applauded, the purpose is not well defined and the complexity of the proposed framework is difficult to follow. 

The full complexity of behavioural motivation is overlooked within the proposed  model and this is a fundamental flaw. 

It might be the the use of language contributes to the lack of clarity in the text, however, I did not find the figures helpful in illustrating the points that were being described within the text. 

Overall, I think that maybe the authors have attempted to achieve too much with this manuscript. Maybe a series of papers the develop sections of the proposed framework would be more helpful, to allow more clarity and depth in defining each component and  illustrating the complexity.

There is some repetition towards the end of section 2.2, a paragraph appears twice. 

I am sorry that I am not able to provide a more positive review, as I think the concept is promising but the manuscript lacks purpose and clarity that preclude it from publication.

Author Response

Dear reviewer:

Thanks very much for taking your time to review this manuscript. We appreciate all your generous comments and suggestions! Please find my itemized responses in below and my revisions in the re-submitted files.

Sincerely yours,

Qikang Zhong, Bo Li, Yue Chen, Jiawei Zhu

Reviewer 3 Report

Please see uploaded file.

Author Response

(The authors gave the same response as above.)
